# Proteostasis Decline and Redox Imbalance in Age-Related Diseases: The Therapeutic Potential of NRF2

**DOI:** 10.3390/biom15010113

**Published:** 2025-01-13

**Authors:** Brigitta Buttari, Antonella Tramutola, Ana I. Rojo, Niki Chondrogianni, Sarmistha Saha, Alessandra Berry, Letizia Giona, Joana P. Miranda, Elisabetta Profumo, Sergio Davinelli, Andreas Daiber, Antonio Cuadrado, Fabio Di Domenico

**Affiliations:** 1Department of Cardiovascular and Endocrine-Metabolic Diseases and Aging, Istituto Superiore di Sanità, 00161 Rome, Italy; brigitta.buttari@iss.it (B.B.); elisabetta.profumo@iss.it (E.P.); 2Department of Biochemical Sciences “A. Rossi Fanelli”, Sapienza University, 00185 Rome, Italy; antonella.tramutola@uniroma1.it; 3Department of Biochemistry, Faculty of Medicine, Autonomous University of Madrid, Centro de Investigación Biomédica en Red Sobre Enfermedades Neurodegenerativas (CIBERNED), National Institute of Health Carlos III (ISCIII), Instituto de Investigación Sanitaria La Paz (IdiPaz), 28049 Madrid, Spain; airojo@iib.uam.es (A.I.R.); antonio.cuadrado@uam.es (A.C.); 4Institute of Chemical Biology, National Hellenic Research Foundation, 116 35 Athens, Greece; nikichon@eie.gr; 5Department of Biotechnology, Institute of Applied Sciences & Humanities, GLA University, Mathura 00185, Uttar Pradesh, India; sarmistha_pharmacol@yahoo.com; 6Center for Behavioral Sciences and Mental Health, Istituto Superiore di Sanità, 00161 Rome, Italy; alessandra.berry@iss.it (A.B.); letizia.giona@iss.it (L.G.); 7PhD Program in Science of Nutrition, Metabolism, Aging and Gender-Related Diseases, Faculty of Medicine and Surgery, Catholic University of the Sacred Heart, 00168 Rome, Italy; 8Research Institute for Medicines (iMed.ULisboa), Faculty of Pharmacy, Universidade de Lisboa, 1649-003 Lisboa, Portugal; jmiranda@ff.ulisboa.pt; 9Department of Medicine and Health Sciences “V. Tiberio”, University of Molise, 86100 Campobasso, Italy; sergio.davinelli@unimol.it; 10Department for Cardiology 1, University Medical Center Mainz, Molecular Cardiology, Johannes Gutenberg University, 55131 Mainz, Germany; daiber@uni-mainz.de; 11DZHK (German Center for Cardiovascular Research), Partner Site Rhine-Main, 55131 Mainz, Germany

**Keywords:** aging, proteostasis, NRF2, neurodegeneration, oxidative stress, redox balance

## Abstract

Nuclear factor erythroid 2-related factor 2 (NRF2) is a master regulator of cellular homeostasis, overseeing the expression of a wide array of genes involved in cytoprotective processes such as antioxidant and proteostasis control, mitochondrial function, inflammation, and the metabolism of lipids and glucose. The accumulation of misfolded proteins triggers the release, stabilization, and nuclear translocation of NRF2, which in turn enhances the expression of critical components of both the proteasomal and lysosomal degradation pathways. This process facilitates the clearance of toxic protein aggregates, thereby actively maintaining cellular proteostasis. As we age, the efficiency of the NRF2 pathway declines due to several factors including increased activity of its repressors, impaired NRF2-mediated antioxidant and cytoprotective gene expression, and potential epigenetic changes, though the precise mechanisms remain unclear. This leads to diminished antioxidant defenses, increased oxidative damage, and exacerbated metabolic dysregulation and inflammation—key contributors to age-related diseases. Given NRF2’s role in mitigating proteotoxic stress, the pharmacological modulation of NRF2 has emerged as a promising therapeutic strategy, even in aged preclinical models. By inducing NRF2, it is possible to mitigate the damaging effects of oxidative stress, metabolic dysfunction, and inflammation, thus reducing protein misfolding. The review highlights NRF2’s therapeutic implications for neurodegenerative diseases and cardiovascular conditions, emphasizing its role in improving proteostasis and redox homeostasis Additionally, it summarizes current research into NRF2 as a therapeutic target, offering hope for innovative treatments to counteract the effects of aging and associated diseases.

## 1. Introduction

The WHO estimates that the number of people aged 80 years or older is expected to triple between 2020 and 2050 to reach 426 million, with Europe having the highest proportion of elderly people in the world (https://www.who.int/news-room/fact-sheets/detail/ageing-and-health and https://ec.europa.eu/eurostat/statistics-explained/index.php?title=Aging_Europe_-_statistics_on_population_developments, accessed on 20 November 2024). Thus, identifying potential molecular hallmarks of aging and developing targeted therapeutic interventions to extend “healthspan” (i.e., the number of years spent in good health), in the face of just extending lifespan, is one of the main challenges of geroscience.

Aging has been defined as a non-adaptive epiphenomenon gradually appearing after the post-reproductive phase of life, leading to a progressive decay in brain and body homeostasis [1]. In this regard, protein and metabolic homeostasis are two regulatory nodes implicated in maintaining cellular and organismal physiology through an extensive synergistic regulation of common targets and cascade signaling. In several model systems of human disease, the protein homeostasis (proteostasis) decline is associated with dysmetabolism, neurodegeneration, and muscle wasting, all pathological conditions mainly present in age-related diseases [2]. However, it is still unclear which mechanism is the cause or consequence of the disease.

The chronic engagement of proteostasis network leads to the low-grade sustained generation of reactive oxygen species (ROS), chronic inflammation, and metabolic imbalance [3]. The regulation of proteostasis via key metabolic pathways/regulators including insulin/IGF1 signaling (IIS), mammalian target of rapamycin (mTOR), 5′ adenosine monophosphate-activated protein kinase (AMPK), and NAD-dependent deacetylases (Sir2-like proteins known as sirtuins) occurs on several levels, reshaping the cellular proteome depending on the cellular energy homeostasis. It is important to realize that disturbances in proteostasis will affect lipid and glucose metabolism and vice versa.

Nuclear factor erythroid 2-related factor 2 (NRF2), as a master regulator of cellular homeostasis, controls the expression of a battery of genes that participate in a broad cytoprotective program, proteostasis, mitochondrial bioenergetics, inflammation, and lipid and glucose metabolism [4]. The accumulation of misfolded proteins influences the release, stabilization, and nuclear localization of NRF2 and in turn, it upregulates the expression of key components of the proteasome and lysosomal degradation machinery, thus actively contributing to the clearance of toxic proteins. Consequently, the pharmacological modulation of NRF2 might mitigate the proteotoxic-derived emergency signals derived from increased oxidative and metabolic stress. In this review, we provide an overview of the NRF2 system and its relationship with the proteostasis network in age-related diseases. We also summarize current studies on NRF2 as a potential target to prevent proteostasis decline and redox imbalance for therapeutic interventions in aging and related diseases [5].

## 2. The Proteostasis Network and NRF2 Response: Molecular Regulation in Health and Disease

### 2.1. Proteostasis Mechanisms: The Balance Between Surveillance and Degradation

Proteostasis denotes the process that controls the dynamic balance of the intracellular pool of functional proteins. A wide and multifaceted network of signaling pathways is adapted according to the proteomic requests of diverse cellular environments and participates in protecting cells and organisms against proteotoxic stress [2]. The careful regulation of the proteostasis network machinery is fundamental in reducing the toxicity associated with the buildup of misfolded and/or damaged proteins. To guarantee proficient folding and avoid aggregation, the cell expresses different types of molecular chaperones that guide nascent polypeptide chains along effective folding pathways [6,7]. The protein-folding networks consist of cytoplasmic and endoplasmic reticulum (ER)-resident chaperones. Chaperones aid proteins in obtaining stable conformations, and in some cases, they impede misfolding and aggregation [8]. Additionally, organelles may retain their own sets of chaperones to guarantee the correct folding to protein that would be needed to reach a different final location [9]. Under pathological conditions, a failure of the chaperone machinery may occur, leading to the buildup of unfolded proteins and ER stress and promoting the induction of the unfolded protein response (UPR) [10,11]. Three types of ER transmembrane anchors—activating transcription factor 6 (ATF6), inositol-requiring protein 1 (IRE1), and PKR-like ER kinase (PERK)—initiate their own signaling pathways upon UPR activation. These pathways regulate the expression of multiple genes involved in the activation of protein quality control systems, redox balance, and protein degradation [12]. Of note, the induction of the PERK branch of the UPR can target NRF2, inducing the expression of proteins involved in the adaptation to oxidative stress (OS) [13]. If mechanisms of ER-resident refolding are not successful, proteins may aggregate and activate clearance pathways that, through the ubiquitin-proteasome system (UPS) and macroautophagy, promote protein degradation. The proteasome employs a refined ubiquitin system to selectively tag and destroy short-lived proteins involved in cell regulation as well as abnormal proteins with irregular structures before they can become harmful to the cell upon deposition [14]. If the attempts of the UPS to clear irregular proteins do not follow through due to cargo overload, defective structures, or excessive ROS, misfolded proteins may accumulate and aggregate [15,16,17]. These aggregates are then targeted for degradation by macroautophagy, which leads to the non-selective lysosomal proteolysis of cytoplasmic constituents [18,19]. Recent data have demonstrated that autophagy, working in tandem with ubiquitin, also performs a selective degradation process [20]. Maintaining proteostasis is critical, particularly in the brain, as central nervous system (CNS) disorders often involve abnormal protein aggregates that, linked to redox imbalance, drive neurodegeneration [21,22]. Neuronal cells rely on antioxidant responses, protein quality control, and degradation pathways, all of which must function effectively to prevent degeneration [23,24].

### 2.2. Proteostasis and Redox Regulation: Is There a Role for NRF2?

The NRF2 transcription factor acts as a key regulator of cellular equilibrium, controlling the expression of over 250 genes under normal and stress conditions through a cis-acting enhancer known as the antioxidant response element (ARE). These genes play roles in various detoxification processes, cellular metabolism, antioxidant defense mechanisms, energy production, lipid metabolism, iron regulation, and the maintenance of protein stability [25].

NRF2 is pivotal in maintaining proteostasis by modulating proteasome subunit expression and upregulating genes essential for the UPS and autophagy, both critical for clearing damaged or misfolded proteins [26]. The UPS marks proteins for proteasomal degradation via ubiquitination, while autophagy encloses them in autophagosomes for lysosomal degradation. NRF2 coordinates the UPS and autophagy, compensating for impairments in either pathway (Figure 1). Supporting this concept, Goldberg and colleagues [27] demonstrated that treating the SH-SY5Y neuroblastoma cell line with a proteasome inhibitor triggered an NRF2-dependent upregulation of autophagy-related genes, such as *p62* and *GABARAPL1*, alongside proteasome subunit genes. NRF2 has been reported to associate with the promoter region of the gene encoding the autophagy adaptor p62, promoting its expression in human cell lines [28]. Moreover, numerous autophagy-related genes possess potential activating ARE sites and are induced through NRF2-dependent mechanisms during oxidative stress (OS) [29].

Cellular proteostasis is supported by a network of chaperones including heat shock proteins like HSP70 and HSP90, which assist in proper protein folding and prevent aggregation. The expression of these chaperones is controlled by heat shock transcription factor 1 (HSF1), which forms trimers and moves to the nucleus under proteotoxic stress [30]. There, HSF1 activates genes encoding molecular chaperones. Notably, proteasome dysfunction, OS, and the heat shock response are closely interconnected [31]. Research has shown that NRF2 can enhance HSF1 transcription during OS in human fibrosarcoma cells [32], and NRF2 overexpression in HEK293T cells increases the RNA levels of chaperones like HSPA4, HSPA8, and HSPA9 [33].

NRF2 also indirectly supports proteostasis by reducing OS. OS can cause protein damage and misfolding, thus, destabilizing the proteome. Through AREs, NRF2 boosts the production of antioxidant enzymes that neutralize ROS, thereby mitigating oxidative damage [34,35]. This protective effect helps maintain a stable and functional proteome, underscoring the integral role of NRF2 in cellular homeostasis and its potential as a therapeutic target for diseases linked to proteostasis imbalance. Redox regulation refers to the control of oxidative and reductive reactions within the cell, maintaining a balance between the production of ROS and their elimination by antioxidant defenses [36]. NRF2 regulates key antioxidant enzymes including SOD, catalase, glutathione peroxidase, and HO-1, which neutralize ROS to protect cells from damage. It also oversees the expression of genes involved in glutathione synthesis, such as GCL, and induces phase II detoxifying enzymes like NQO1 to eliminate reactive metabolites and xenobiotics [35]. Proteostasis and redox regulation are interconnected processes. OS can disrupt proteostasis by damaging proteins, while the accumulation of damaged proteins can exacerbate OS. NRF2 serves as a key mediator that integrates these two processes [13].

Organisms are continually exposed to exogenous and endogenous sources of ROS and other oxidants that have both beneficial and deleterious effects on the cell. ROS have important roles in a wide range of physiological processes; however, high ROS levels are associated with OS in various diseases including conditions like Alzheimer’s and Parkinson’s disease, cardiovascular diseases such as atherosclerosis, metabolic disorders like diabetes, and cancer. In these conditions, elevated ROS levels contribute to cellular damage, inflammation, and tissue dysfunction, all of which accelerate disease progression [37,38].

### 2.3. NRF2 Activation in Response to Proteotoxic-Derived Emergency Signals

Secretory and membrane-bound proteins are synthesized in the rough ER through a regulated process that ensures only properly folded proteins reach the Golgi. Protein folding involves forming disulfide bonds, a process driven by enzymes like protein disulfide isomerase (PDI) and ERO1. However, this process also generates hydrogen peroxide, which is managed by protective enzymes to prevent damage. If the system fails, misfolded proteins build up in the ER, triggering the UPR, which works to restore normal function [39]. When this redox system malfunctions, misfolded proteins accumulate in the ER, triggering the UPR. Under normal conditions, PERK, IRE1, and ATF6 are kept inactive by the chaperone GRP78/BIP. During ER stress, GRP78 binds to unfolded proteins, releasing and activating the indicated sensors and triggering a response involving XBP1, ATF4, and cleaved ATF6 transcription factors that help restore cellular balance [40,41]. The accumulation and aggregation of misfolded proteins induce excessive ROS production from mitochondria, the ER, and other sources, which can activate NRF2 [4]. This mechanism has been thoroughly studied in Caenorhabditis elegans, where skinhead-1 (SKN-1), the NRF2 ortholog, transactivates a protective transcriptional program. Under ER stress conditions, SKN-1 directly activates many genes involved in ER function including canonical ER signaling and transcription factors that in turn induce skn-1 transcription. Importantly, this response is distinct from its response to OS. SKN-1 is required for resistance to ER stress including reductive stress, a surprising finding given its established role in OS defense. Interestingly, UPR signaling is necessary for SKN-1 to mobilize an OS response, suggesting that the ER has a licensing and possibly sensing role during oxidative and xenobiotic stress responses [42]. Mechanistically, PERK-induced stress phosphorylates NRF2, leading to its stabilization (Figure 1).

In PERK-null cells, apoptosis is linked to peroxide buildup and impaired protein synthesis. PERK activates NRF2 by phosphorylating it at Ser40, preventing its degradation by KEAP1 [43]. The IRE1-TRAF2-ASK1 pathway may also activate NRF2, though the role of MAPKs remains debated [44,45]. Recent studies show that cysteine sulfenylation of IRE1 inhibits UPR but triggers NRF2-driven antioxidant responses [46]. In keratinocytes, IRE1α loss reduces PERK-mediated NRF2 activation, lowering antioxidant gene expression after UVB exposure. These cells show high basal ROS levels but fail to induce ROS upon UVB exposure [47]. SMURF1 overexpression promotes NRF2 activity by degrading KEAP1, reducing ROS, and mitigating ER stress-induced cell death [48]. NRF2 can also be degraded independently of KEAP1 via the E3 ligase HRD1, a process modulated by Pak2 in response to UPR inducers like tunicamycin [49,50]. A positive feedback loop exists between NRF2 and UPR: NRF2 activates ATF4, and together, they induce Hmox1 expression to enhance antioxidant defenses [44,51]. In addition, NRF2 has emerged as a critical regulator in preventing the shift from a pro-survival to a pro-death response when homeostasis fails to be restored. Bioinformatic analysis, followed by experimental validation, has identified NRF2 as a key factor in mitigating ER stress-induced pyroptosis in the murine hippocampus [52]. Olfactory neurons subjected to systemic tunicamycin administration showed increased NRF2 expression alongside other UPR members such as CHOP, BIP, and XBP1 [53]. These findings have been corroborated by in vivo studies, where the lateral ventricular infusion of tunicamycin in rats induced PERK and NRF2 expression in the hippocampus, accompanied by significant cognitive deficits, increased TAU phosphorylation, and Aβ42 deposition [54]. Similarly, NRF2 activation in zebrafish alleviated ER stress caused by a mutation in the phosphomannomutase 2 gene, which affects early N-glycosylation steps [55]. A proof-of-concept study in this model demonstrated that activation of the ER chaperone sigma-1 receptor mitigates mutant TDP43 toxicity by boosting the ER stress response and antioxidant defense through NRF2 signaling [56].

Conversely, NRF2 knockout cells exhibit increased cell death compared to wild-type cells when exposed to tunicamycin [43], and its activation depends on KEAP1 degradation by p62/SQSTM1 [57]. In line with this, shRNA-mediated silencing of NRF2 in βTC-6 cells, a murine insulinoma β-cell line, significantly increased tunicamycin-induced cytotoxicity, elevated the expression of the pro-apoptotic ER stress marker CHOP10, and inhibited tunicamycin-inducible expression of the proteasomal catalytic subunits Psmb5 and Psmb6. Similarly, NRF2 silencing abolished the protective effects of 1,2-dithiole-3-thione (D3T) against ER stress. These results suggest that NRF2 contributes to the ER stress response in pancreatic β-cells by enhancing proteasome-mediated ERAD [58]. In tunicamycin-induced ER stress and liver injury models, the NRF2 levels were closely correlated with SIRT3. NRF2 deficiency increased tunicamycin-induced CHOP expression, which was mitigated by SIRT3 overexpression [59].

In summary, the activation of NRF2 by the UPR helps protect cells from ER stress by reducing oxidative damage and enhancing protein degradation. Dysregulation of this pathway can lead to increased OS, protein aggregation, and cell death, with implications for neurodegenerative diseases and other conditions associated with protein misfolding (Figure 2).

## 3. Aging and Longevity

### 3.1. The Effects of NRF2 Modulation on Aging, Healthspan, and Longevity

Aging is a multifaceted, inevitable natural process characterized by increasing damage in all cellular molecules (e.g., DNA, RNA, proteins, lipids, etc.) and consequently, downstream alterations in all metabolic pathways. This, in turn, constantly enhances frailty and increases death probability [60]. Aging also represents a fundamental risk factor for the development of age-related diseases including neurodegenerative diseases [61]. More recently, the importance of controlling frailty via maintenance of the cellular/tissular/organismal well-being for longer periods has emerged, and the notion of healthspan maintenance is recognized, even in the absence of lifespan extension [62].

Among the hallmarks of aging, one can find cellular senescence, loss of proteostasis and disabled macroautophagy [63]. Even if one takes into consideration only those three hallmarks, the emerging implication of NRF2 is evident. More specifically, premature cellular senescence occurs upon the knockdown or inhibition of NRF2 in human primary HFL-1 fibroblasts [64,65] and endothelial progenitor cells [66]. In contrast, its timed activation enhances cellular longevity [65]. Interestingly, prolonged NRF2 activation promotes fibroblast senescence and induces a cancer-associated fibroblast phenotype [67], thus suggesting that the duration and magnitude of NRF2 activation are critical for the outcome on aging and longevity. Concerning proteostasis, various proteasome genes have been revealed to be coordinately regulated by NRF2 modulators [68], while interestingly, the observed lifespan extension following genetic proteasome activation is dependent, among others, on the NRF2 functional ortholog SKN-1 in *C. elegans* [68]. Likewise, NRF2 has been identified as the regulator of several macroautophagy genes [29]. Therefore, upon NRF2 modulation either through specific compounds or through genetic manipulation, effects on aging, healthspan, and longevity are expected to occur. The list of signals or pharmacological agents that increase the lifespan of various organismal models via the NRF2 pathway is too long. Therefore, we will only refer to studies where the genetic manipulation of NRF2 (or its negative regulator, KEAP1) has been performed and the effects on aging, healthspan, and longevity have been reported.

Work in lower eukaryotes has revealed the link between NRF2, aging, and longevity. SKN-1 has been shown to play an instrumental role in longevity in *C. elegans*. Loss-of-function *skn-1* mutants are short-lived [69]. Although the expression of SKN-1 from high-copy arrays is toxic, modest SKN-1 overexpression enhances longevity [70]. On top of that, maintenance of the movement for longer occurs upon modest SKN-1 overexpression, thus suggesting a better healthspan of the animals. Enhanced CncC/NRF2 expression in *Drosophila melanogaster* through *keap1* loss-of-function mutations extends the lifespan [71]. Likewise, silencing of KEAP1 has beneficial effects on synaptic function and longevity [72,73]. CncC silencing reduces negative geotaxis activity while KEAP1 silencing increases negative geotaxis activity, thus suggesting NRF2-mediated effects on healthspan [73]. The effects of ubiquitous overexpression of CncC/NRF2 are dependent on the dose in the fly in accordance with what was reported in the worm. More specifically, mild CncC/NRF2 activation extends lifespan, while high CncC/NRF2 expression levels result in developmental lethality. It is, however, noteworthy that persistent activation of CncC/NRF2 in adult flies results in accelerated aging [72]. Nevertheless, pan-neuronal overexpression of CncC does not affect longevity [73]. Work has been also performed in mammals. Rodents with enhanced NRF2 activity live longer than rodents with low activity [74,75]. The phenomenal lifespan of naked mole rats has been at least partially correlated with an enhanced NRF2 status [75]. Upon NRF2 deletion in C57BL/6J mice, a shortened lifespan has been reported under ad libitum conditions [76], whereas the caloric restriction-mediated pro-longevity is NRF2-independent [77]. Mice with reduced *KEAP1* expression (allowing a moderate NRF2 activation) display ameliorated healthspan features like attenuated age-related hearing loss [78] or delayed age-dependent deterioration of the salivary glands [79]. Unfortunately, none of these studies reported anything on the lifespan curves of those animals.

Little is known about the role of NRF2 in human aging per se. NRF2 impairment is suggested to be a key player in Hutchinson–Gilford progeria syndrome (HGPS), a rare premature aging disorder. Repressed NRF2 transcriptional activity is found in HGPS fibroblasts, whereas NRF2 reactivation reverses the progerin-induced aging defects [80]. An enhanced NRF2 activity is detected in centenarians, possibly linked to their dietary habits enriched in phytochemical/NRF2 inducers [81], but more experiments are needed. While NRF2 has been identified as a potential key regulator of aging, additional research is required to fully understand its role. Any attempts to activate NRF2 should be carefully controlled, as its positive effects are closely dependent on both timing and dosage.

### 3.2. NRF2 and Oxidative Stress from Early Developmental Stages to Senescence: A Gender Perspective

The aging process is characterized by a variety of physiological changes involving the decrease in antioxidant defenses (including reduced expression levels of NRF2), increased OS, and the development of a chronic low-grade inflammatory state affecting cellular homeostasis and proteostasis [82,83]. Such condition, referred to as “inflammaging” [84], is common to non-communicable age-related pathologies such as type 2 diabetes mellitus (T2D), cardiovascular diseases (CVDs), and neurodegenerative diseases that frequently coexist in the same aged individual and mutually reinforce the overall decreasing healthspan. Both genetic and environmental factors that might contribute to set vulnerability to unhealthy aging and early life adversities have been widely reported to play a role. Mammals are characterized by an elevated degree of plasticity during ontogenetic phases, and early prenatal and postnatal phases are extremely sensitive to the environment that—with respect to the genetic background—contributes to shaping developmental trajectories [85]. Thus, negative or stressful events experienced in utero, or soon after birth, might program the offspring’s cells, tissues, organs, structure, and function, ultimately setting the stage for health outcomes during aging [85,86,87,88]. We have recently provided evidence for the long-term consequences of NRF2 early programming by prenatal stress [89]. In particular, we have shown that both maternal obesity as well as maternal stress increased OS and inflammation in the female offspring brain, also affecting emotional reactivity. These changes, observed in the hippocampus, were associated with decreased levels of NRF2 and brain-derived neurotrophic factor (BDNF), a neurotrophin that plays a role in brain plasticity [90]. These data suggest that prenatal stress affects the NRF2-dependent network toward a reduced ability to cope with proteotoxic insults. Moreover, it also suggests that early-life adversities might lead to an increased vulnerability to age-related disorders through epigenetic mechanisms involving the ability to maintain cellular and protein homeostasis [91]. N-Acetylcysteine (NAC) is a powerful antioxidant and a precursor of glutathione (the major thiol redox buffer to maintain intracellular redox homeostasis); it is a well-tolerated and widely used compound in clinical practice. NAC administration to pregnant dams was able to protect the offspring from the negative effects of prenatal stressors by modulating glutathione levels in the brain and rebalancing glucose homeostasis in adult mice [92]. Moreover, it increased the NRF2 expression levels in the brain of adolescent subjects. Once again, these effects were sex-dependent and suggest that NRF2 plays a central role in the response to prenatal OS-mediated insults that might hold long-term consequences [90].

Sex differences in the aging process, and in the ability to cope with oxidative and proteotoxic stress and inflammatory insults, are well-known in geroscience. Indeed, evidence from both animal and human studies shows that the male sex is characterized by increased ROS production and lower antioxidant defenses than females, a difference that becomes less pronounced after menopause [93,94,95,96]. Nevertheless, the reduced exposure to OS insults over the life period preceding old age may protect females, resulting in a longer lifespan. In contrast, although males are at higher risk of morbidity and mortality following an acute inflammatory process, females show the worst prognosis in chronic conditions and are at higher risk of the development of autoimmune and neurodegenerative diseases [97]. Thus, sex/gender differences should be carefully considered in the attempt to devise target-specific therapeutic interventions to promote healthy aging. Natural compounds are becoming increasingly popular in anti-aging research for their safety profile and pleiotropic effects [98,99,100]. One of the main features characterizing these compounds is their ability to boost antioxidant and anti-inflammatory defenses through different mechanisms also involving the modulation of the NRF2 machinery [101,102,103]. In this regard, we have shown that the administration of trehalose (a disaccharide of glucose synthesized by fungi, plants, and invertebrates) in senescent mice results in sex-dependent effects, leading to a healthier phenotype. In particular, trehalose-treated males showed improved motor learning and better performance in coordination tasks. This behavioral phenotype was associated with the activation of the UPS, autophagy, and antioxidant defenses in the cerebral cortex. In contrast, and independently from trehalose administration, females showed better motor performance in association with higher levels of ubiquitinated proteins and NRF2 in the cerebral cortex, strengthening the body of evidence in support of a female advantage concerning antioxidant defenses [104]. Such an effect might be explained at least in part through the antioxidant and immunomodulating properties of estrogens that might overall prime the female physiology with protective effects still visible during the post-reproductive phase. This hypothesis is also in line with our previous data showing preserved neurogenesis only in the brain of old female mice, an effect potentiated by the genetic reduction in OS (p66Shc−/− mice) [105]. Thus, NRF2 appears to play a key role in stress responsiveness at all life stages. Starting from prenatal life, it may affect fetal programming and, in the long-term, may modify health trajectories, making the individual vulnerable to late-onset non-communicable diseases with the main consequences on healthspan during aging.

An ever-increasing body of evidence suggests that the optimal balance between pro-oxidant and antioxidant species needs to be constantly and dynamically adjusted to the specific lifetimes and physical and environmental challenges including pollution, diet, physical activity, and toxin exposure. These factors can influence oxidative stress, either exacerbating it or helping to maintain the pro-oxidant/antioxidant. For example, a moderate increase in ROS and inflammation is crucial to promote embryogenesis, fetal development, parturition, or to promptly counteract infections. In contrast, a reduction in inflammation and OS is associated with increased longevity and improved healthspan. Interestingly H_2_O_2_ is specifically involved in the promotion of fat accumulation within the adipocytes by reinforcing insulin signaling. Such a condition, which is detrimental in Westernized countries where high caloric foods poor in nutrients are constantly available, is life-saving for organisms living in harsh conditions with scarce food availability. From this point of view, the cellular machinery responsible for the maintenance of proteostasis, which includes NRF2-KEAP1 crosstalk, should also adapt by tightly modulating the expression levels to ensure reduced cellular stress and promote survival [87,106].

## 4. Cellular Dysfunctions in Age-Related Diseases and the Therapeutic Potential of NRF2

### 4.1. The Interplay Between Antioxidant Responses and Proteostasis in Alzheimer’s Disease, Alzheimer-like Dementia, and Parkinson’s Disease

Alzheimer’s disease (AD) is a progressive neurodegenerative disorder marked by cognitive decline and memory loss, primarily affecting the elderly population. Pathologically, AD is characterized by the accumulation of amyloid-beta (Aβ) plaques and neurofibrillary tangles (NFTs) composed of hyperphosphorylated tau protein. OS is a significant contributor to the pathogenesis of AD, with increased levels of ROS causing damage to proteins, lipids, and DNA [107,108]. Several mechanisms have been proposed to explain the onset of ROS production in the brain of individuals with AD.

Under physiological conditions, elevated ROS levels prompt cells to produce antioxidants to prevent damage. The transcription factor NRF2, which regulates the expression of antioxidant genes (i.e., *HO-1*, *NQO1*, *GCLC,* and *GCLM*) in response to OS, plays a key role in this process.

However, in AD, the NRF2 levels are reduced, impairing the activation of antioxidant pathways [109]. In the hippocampus of AD patients, NRF2 is predominantly cytoplasmic, limiting its function as a transcription factor for antioxidant genes [109]. Age-related declines in Nfe2l2 mRNA in non-transgenic mice suggest a link between aging and oxidative damage in AD [110]. Similarly, APP/PS1 mice show reduced NRF2 levels and decreased expression of its antioxidant targets including NQO1, GCLC, and GCLM [111]. Conversely, enhancing NRF2 activity in transgenic mice reduces oxidative damage and improves cognitive function [112]. NRF2 deficiency worsens AD pathology. NRF2 knockout mice show increased Aβ deposition, neuronal loss, OS, and neuroinflammation. Mice lacking NRF2 in a model combining amyloidopathy and tauopathy exhibit elevated phosphorylated-TAU and Aβ *56 levels, along with deficits in spatial memory and synaptic plasticity [112,113]. Additionally, genetic variations in the NRF2 gene (*NFE2L2*) influence susceptibility to AD, with some polymorphisms linked to altered NRF2 activity and OS response, potentially reducing the risk of neurodegeneration [114].

In light of this evidence, NRF2 may have a protective role in AD, making it a potential target for therapeutic intervention. The pharmacological activation of NRF2 represents a promising therapeutic approach for AD. Compounds such as sulforaphane (SFN), found in cruciferous vegetables, and dimethyl fumarate, an FDA-approved drug for multiple sclerosis, have been shown to activate NRF2 [115,116]. In preclinical models of AD, these compounds reduce Aβ-accumulation, improve synaptic plasticity, and enhance cognitive function [115,116]. In the last decade, a major attempt has been made to identify compounds that can increase NRF2 signaling. Several compounds have been shown to activate the NRF2 pathway and attenuate AD-related pathology in animal models. Rosmarinic acid (RosA) activates NRF2 through the protein kinase B/GSK3β pathway, reducing Aβ-induced OS [117]. Carnosic acid (CA), found in rosemary and sage, reduces Aβ levels and inflammation in mouse models of AD, though its effects on cognition have been mixed [118]. Mini-GAGR and gracilin A, both of which enhance NRF2 activity, have shown reductions in Aβ and tau pathology, though their impact on cognitive function is variable [119]. Other natural products like hydrogen sulfide, forsythoside, and anthocyanin also enhance NRF2 signaling, leading to reduced Aβ and tau accumulation and improved cognitive outcomes in AD models [120,121,122]. Given the promising results from animal studies, various strategies to enhance NRF2 function have been explored in AD models, with many yielding positive outcomes [123]. Currently, several clinical trials are underway to test small-molecule drugs that activate NRF2 in AD patients (e.g., NCT02292238; NCT02711683; NCT02085265; NCT04213391). Early findings from these trials show some encouraging results [124].

Down syndrome (DS), caused by trisomy of chromosome 21 (HSA21), is strongly associated with Alzheimer’s-like dementia, characterized by amyloid plaques and neurofibrillary tangles [125]. DS is increasingly linked to disrupted redox homeostasis, with OS playing a key role [126,127]. The triplication of HSA21 leads to the overexpression of genes like *SOD1* and *BACH1*, contributing to elevated ROS. Overexpressed SOD1 increases H_2_O_2_ levels, which are insufficiently neutralized, causing oxidative damage [128]. BACH1, a transcriptional repressor competing with NRF2, is overexpressed in DS brains, reducing NRF2 activity and antioxidant defenses, thereby exacerbating OS and promoting AD pathology [129,130]. Similarly to what has been observed in AD, the depletion of the NRF2 response, observed in the DS human brain, contributes to neuropathological mechanisms, but at an earlier time point due to BACH1 triplication [129]. Analogous findings regarding the NRF2/BACH1 ratio have been obtained in DS mice and peripheral blood mononuclear cells (PBMCs) derived from children with DS and in mouse models of the disease [131,132]. Conversely, Zamponi and collaborators reported NRF2 activation in human astrocytes and fibroblasts from individuals with DS [133]. Intriguingly, UPR amelioration was accompanied by the rescue of NRF2/BACH1 balance and the reduction in protein oxidation. Downstream of the UPR, the analysis of proteasome degradation functionality in the brains of individuals with DS showed a decrease in trypsin-like, chymotrypsin-like, and caspase-like activities, indicating impaired protein clearance during the early stages of the disease [134]. Aberrant mTOR/autophagy signaling is an early degenerative event in DS brains, contributing to Alzheimer’s-like cognitive decline. Hyperactivation of the PI3K/AKT/mTOR pathway is linked to reduced autophagosome formation and elevated Aβ and p-tau levels [135,136,137,138]. Studies also highlight endocytic alterations and suppressed macroautophagy in DS [139,140]. Notably, rapamycin, an mTOR inhibitor, improves cognitive performance, reduces APP and tau pathology, and mitigates oxidative stress in DS mouse models [136,138].

Regarding cancer, people with DS are at similar or slightly lower overall risk compared to the general population [141]. In addition, patients with DS display a unique spectrum of malignancies, with a 10- to 20-fold increased risk of both acute lymphoblastic and myeloblastic leukemia (especially megakaryoblastic acute leukemia) [142]. On the contrary, solid tumors (e.g., breast cancer, prostate cancer, medulloblastoma, neuroblastoma and Wilms tumor) are markedly less frequent in DS individuals across all age groups, except for retinoblastoma and germ cell tumors [141]. However, given the increased life expectancy of individuals with DS in recent decades, the risk of cancer in adulthood is becoming increasingly important. Studies have shown that NRF2 activation can contribute to tumor development and progression by promoting cell survival, proliferation, and resistance to chemotherapy and radiation therapy [143]. Normally, NRF2 is degraded through a Keap1-Cul3-Roc1-dependent mechanism, but in human cancer, somatic mutations occur in NRF2, resulting in its accumulation. Moreover, NRF2 activation can increase the anti-apoptotic gene expression and promote carcinogen metabolism, leading to cell survival and increased DNA damage and mutation rates, which can further promote cancer development [144]. No specific studies have disclosed the role of NRF2 in cancer development in people with DS, however, the overexpression of BACH1, associated with the mitigation of NRF2 induction and the pro-apoptotic phenotype, might address the reduced risk of developing solid tumors.

Although AD and Parkinson’s disease (PD) are characterized by distinct clinical features and pathological hallmarks, they share the abnormal accumulation and misfolding of proteins [145]. Alpha-synuclein (α-syn) aggregates into Lewy bodies, which play a central role in PD and contribute to dopaminergic neuronal death, synaptic dysfunction, and neurodegeneration in the substantia nigra. It has been reported that NRF2 mitigates neurodegeneration induced by α-syn in PD by enhancing protein degradation and maintaining proteostasis. This effect reduces the toxicity of misfolded proteins, decreases α-syn levels, and improves neuronal survival [146,147].

NRF2 has also emerged as a key regulator of proteasome homeostasis, particularly in the context of PD. Genetic deletion of NRF2 impairs proteasome activity, which in turn aggravates α-syn toxicity and exacerbates both the degeneration of nigral dopaminergic neurons and neuroinflammation [146]. Notably, the upregulation of the many 20S proteasome subunits, which contains ARE sequences in its 5′-untranslated regions, are mediated by NRF2, enhancing proteolytic capacity and stress resistance [148]. Furthermore, DJ-1, a small protein ubiquitously found in almost all tissues including the brain, regulates the activity of the 20S proteasome and plays a critical role in the pathogenesis of PD through its regulation of the OS response. Under conditions of OS, DJ-1 stabilizes NRF2 activity by inhibiting its interaction with KEAP1, thereby promoting the upregulation of both the 20S proteasome and NQO1, which ensures the degradation of oxidatively damaged proteins [149].

Dysregulation of PTEN-induced putative kinase 1 (PINK1) and Parkin, a serine/threonine kinase and an E3 ubiquitin ligase, respectively, is one of the most common causes of PD. These proteins are key players in mitophagy, a selective form of autophagy that eliminates dysfunctional mitochondria. It has been demonstrated that the loss of function of PINK1 and Parkin results in the accumulation of damaged mitochondria within neurons, leading to increased ROS levels and the subsequent death of dopaminergic neurons [150]. Recent findings indicate that NRF2 can regulate several aspects of mitophagy, both through the traditional autophagy receptor p62 and by reversing the effects of Parkin/Pink1 knockdown in a neuromuscular degenerative model of transgenic flies. Specifically, when PINK1 and Parkin are knocked down in neuronal tissues, the activation of NRF2 induces proteostatic pathways, reduces OS, and increases mitophagy in dopaminergic neurons [151].

All of these results indicate that NRF2 may be a druggable target in PD. For example, using an animal model of PD, it has recently been demonstrated that dimethyl fumarate exerts neuroprotective effects by enhancing mitophagy through the NRF2/PINK1 axis, together with BCL2 interacting protein 3 (BNIP3), another key protein involved in the regulation of mitophagy [152]. Dimethyl fumarate has been shown to reduce α-syn aggregates and attenuate neuronal cell death in mouse models of PD [153]. Furthermore, a considerable number of natural compounds targeting NRF2, such as epigallocatechin-3-gallate (EGCG) and curcumin, have been found to inhibit the aggregation of α-syn in both in vitro and in vivo models of PD [154,155,156]. Although many NRF2 activators have low bioavailability and induce off-target effects, a literature review by Niu et al. highlighted the efficacy of various NRF2 inducers, such as SFN, ellagic acid, and caffeic acid, among others, in improving proteostasis and motor function in PD models through the regulation of NRF2 activity [157]. Recently, a dietary supplement containing tocotrienols derived from palm oil, commercially known as Tocovid SupraBio (Hovid Inc., Ipoh, Malaysia), has moved into a phase II clinical trial for PD [158]. Tocotrienols, which can activate NRF2 and its target genes, have been shown to prevent neurotoxicity and motor deficits in a mouse model of PD [159,160]. Sesaminol, a lignan derived from sesame seeds, has demonstrated neuroprotective effects against OS-induced apoptosis in SH-SY5Y cells, which exhibit dopaminergic-like properties, by promoting the nuclear translocation of NRF2. The same study revealed that sesaminol reduces the expression of α-syn in the substantia nigra of the brain in a rotenone-induced PD model [161].

### 4.2. Proteostasis and NRF2 Regulation in Cardiovascular Diseases

Impaired proteostasis and the accumulation of protein aggregates are hallmarks of the aging process [3] and age-associated cardiovascular disorders such as atherosclerotic cardiovascular disease, cardiac hypertrophy, cardiomyopathies, and heart failure [162]. The prevalence of CVD markedly rises with age, but risk factors such as hypertension, type 2 diabetes, dyslipidemia, obesity, comorbidities, and other risk factors such as tobacco abuse, alcohol abuse [163,164], and air pollution [165] contribute to the activation of OS, the accumulation of proinflammatory cytokines, endothelial dysfunction, and insulin resistance/hyperinsulinemia [166].

The decline in proteostasis and the presence of OS are highly implicated in CVD [167,168,169], raising the possibility of utilizing NRF2 as a therapeutic intervention. Thus, identifying the bridges that connect proteostasis and the NRF2 pathway could be extremely beneficial to novel dual-target therapies.

Inflammation and OS are known to impair the folding of protein and protein quality control by inducing post-translational modifications [170] or by enhancing protein amyloidogenic propensity or misfolded protein load [171]. In aging and age-related chronic diseases, disturbances in the protein quality control machinery, which includes molecular chaperones, the UPS, and autophagy/lysosome pathway [172], have been observed to be associated with mitochondrial dysfunction, impaired protein homeostasis (proteostasis) network, and alteration in the activities of transcription factors such as NRF2 and NF-κB.

Dysregulation of proteostasis plays a crucial role in the development and progression of atherosclerosis, a hyperlipidemia condition characterized by the buildup of plaque in the arteries, which can lead to serious health complications such as heart attack or stroke. In atherosclerotic plaques from mice and humans, atherogenesis arises from an initial event destabilizing the low-density lipoprotein (LDL) structure and conformation. The accumulation in the vascular subendothelial space of modified LDL leads to the accumulation of cytotoxic, pro-inflammatory, and protease-resistant aggregates similar to other well-known protein deposition diseases [173]. The accumulation of lipoprotein-derived amyloid in atherosclerotic lesions is a result of the reduced conformational stability of lipoproteins due to enzymatic or oxidative modifications as well as the absence or reduction of lipids [174]. Intracellular amyloid deposits can be released following endothelial cell (EC) death, and as a result can form extracellular amyloid deposits, where they can activate and enhance platelet aggregation and degranulation, stimulating the formation of new atherosclerotic plaques or cause direct damage because of cytotoxicity [175,176,177]. Since NRF2 serves as a repressor of KEAP1, an imbalance in the NRF2:KEAP1 ratio, driven by the age-dependent differential expression of NRF2 and KEAP1 [178,179], may contribute to the loss of proteostasis in vascular cells. Notably, elevated levels of free and S-nitrosylated KEAP1 have been observed in aged ECs, where they act as critical, independent regulators of EC function and protein S-nitrosylation [178]. Aged ECs showed the deposition of protein aggregates, dysregulated proteostasis, and impaired autophagy, and a similar EC phenotype was observed in cells from young NRF2 knockout mice. Accordingly, pharmacological NRF2 activation may prevent EC premature aging and senescence, thereby preserving the important endothelial function to confer vasodilation and suppress platelet aggregation and the adherence of leukocytes to the vascular wall [180].

Impaired proteostasis, inflammation, and OS are pathological hallmarks of heart failure, a complex syndrome associated with reduced cardiac function, cardiac remodeling, and cardiomyocyte cell death [181,182,183,184,185]. At the cellular level, decreased contractility in heart failure has been linked to the functional decline of the sarcomere, the fundamental molecular unit of contraction, caused by mechanical stress and OS that predispose proteins to misfold and contribute to the deterioration of the protein quality control system [186]. In cardiac amyloidosis, an important cause of heart failure, the deposition of misfolded immunoglobulin light chains and transthyretin, either in its wild-type or mutated form [187,188], leads to cardiotoxic damage of atrial cardiomyocytes and subsequent cardiac dysfunction. As a maladaptive response to atrial fibrillation, the human atrium produces amyloid, mainly composed of the atrial natriuretic type and to a lesser degree, the B-type natriuretic peptide [189]. Both aging and chronic hypertension lead to the accumulation of aggregates in the heart, and these aggregates may resemble the aggregates that occur in the cortex of AD patients, thus indicating common underlying mechanisms in the deterioration of the proteostasis-maintenance machinery [190].

Several studies have provided strong evidence that amyloid fibrils or amyloidogenic species can be harmful to cells and their recognition by pattern recognition receptors such as the receptor for advanced glycation end-product, scavenger receptors, and toll-like receptors [191,192,193] may amplify the chronic inflammation associated with CVD.

Molecular chaperones are expressed at high levels in the cardiomyocytes and are responsible for the correct folding and assembly of the sarcomeric proteins desmin, myosin, and actin [194,195]. Different studies have demonstrated the cardioprotective effects of chaperones in CVD, particularly ischemia-reperfusion (I/R) injury [196], heart failure [197], and atrial fibrillation [198,199], thus representing a possible therapeutic target. Chaperones can effectively counteract the toxicity of aberrant protein aggregates through their ability to interact with them by promoting their disassembly or simply engaging with them. The overexpression and induction of molecular chaperones are protective in a wide range of animal models of amyloidosis [198,200,201]. Therefore, developing methods to enhance the potential of molecular chaperones to disaggregate proteins and revert protein aggregation has emerged as an appealing strategy for delaying the onset of pathologies related to protein aggregation.

As described above (see Section 2.2), various studies have identified electrophilic compounds that are able to activate both NRF2 and HSF1 by binding to AREs and heat-shock factor response elements (HSE), respectively, and activating the production of HSPs, thus protecting cells against OS [202,203,204]. An experimental study on a mouse model of combined hyperglycemia and hyperlipidemia showed that the administration of 17-dimethylaminoethylamino-17-demethoxygeldanamycin (17-DMAG) reduced the expression of proinflammatory and profibrotic genes, improved renal function, reduced atherosclerotic lesions, and promoted a more stable plaque phenotype [205]. The ability of 17-DMAG to improve nephropathy and atherosclerosis in diabetic mice has been ascribed to the inhibition of HSP90 and to the induction of the protective HSP70, accompanied by the inactivation of NF-kB and inhibition of the signal transducers and activators of transcription (STAT) as well as target gene expression [205]. Despite the protective effects of HSPs, in certain conditions, mutations of chaperone proteins or alterations of their expression can have a deleterious effect on diseases depending on the correct function of the proteostasis machinery. In a previous study, we demonstrated that exposure of human umbilical vein EC (HUVEC) to a pro-oxidant microenvironment upregulated HSP90 surface expression in these cells, suggesting that the oxidative microenvironment of atherosclerotic plaques promotes the upregulation of HSP90 expression on the surface of endothelial cells, thus rendering the protein a possible target of autoimmune reactions [206,207,208]. Atherosclerosis risk factors such as inflammation and OS also promote surface HSP60 expression in vascular endothelial cells. The autoimmune reactivity induced against HSP60 significantly damages endothelial cells and contributes to the development of atherosclerosis [209,210,211]. Therefore, inhibiting excessive OS also counteracts the progression of CVD by regulating the expression of HSPs. Of interest, data obtained on streptozotocin-induced diabetic apolipoprotein E-deficient mice treated with the HSP90 inhibitor 17-dimethylaminoethylamino-17-demethoxygeldanamycin demonstrated that HSP90 inhibition promoted the activation of NRF2 in the aortic tissue, and that NRF2 induction was associated with a concomitant inhibition of NF-κB in the atherosclerotic plaques, thus determining the reduction in the lesion size and inflammation [212].

It is emerging that NRF2 shares some transcriptional targets with HSF1 and that these two transcription factors compensate each other in the regulation of the cellular redox balance [213]. In a more recent study carried out on human cells exposed to arsenite treatment to induce oxidative stress, two AREs were identified in the *HSF1* gene promoter, and it has been demonstrated that NRF2 interacts with these regions and forms a multiprotein activation complex with the chromatin modifier BRG1, thus promoting *HSF1* gene expression [214]. Transient upregulation of HSF1 and NRF2 promotes the reduced state of cells, thus playing an important role in cytoprotection and indicating these two factors as attractive therapeutic targets.

Autophagy dysfunction is another important mechanism implicated in CVDs. Autophagy activity is usually reduced by age [215]. Exhaustion of the aged autophagic machinery is linked to a hyper-activation of the nutrient response pathway defined by the mTOR and a reduction in the activity of the sensor of energy status AMPK, which subsequently inactivate the pro-autophagic ULK1 complex [216]. In atherosclerotic lesions of the aortal intima, dysfunction of various cytosolic and mitochondrion-localized proteins involved in autophagy regulation contributes to abnormal mitochondrial turnover and the elimination of damaged mitochondria and misfolded protein [217,218].

In the early stages of atherosclerotic plaques, the levels of autophagy chaperone p62/SQSTM1 protein and ubiquitin are significantly reduced. In contrast, the levels of p62 and ubiquitin are significantly increased in the late stage of atherosclerosis, especially in the lipid-rich region, indicating a dysregulation of the autophagy pathway. Several studies suggest that when autophagy is activated, p62 plays an important role in stabilizing the atherosclerotic plaque. When the autophagic activity is diminished due to the physiological age-related alterations in the proteostasis, p62 promotes the instability of the plaque. Lipid oxidation, inflammation, and metabolic stress may impair autophagy and in particular affect the degradation of triacylglycerol and cholesterol, contributing to hyperlipidemia in atherosclerosis. In addition, LDL mimicking insulin action can suppress autophagy through activation of the PI3K/Akt/mTOR signaling pathway in endothelial cells [219]. Additionally, while basal and mild adaptive autophagy play a protective role against the progression of atherosclerotic plaques [220,221], excessive activation of autophagy gene expression in response to vascular inflammation may result in autophagy-dependent cell death of all three types of cells—EC, vascular smooth muscle cells, and monocytes/macrophages—that participate in the development of plaque [222,223]. Selective inhibition of the Akt/mTOR signal pathway stabilized the vulnerable atherosclerotic plaques by promoting autophagy, also by targeting the NRF2 pathways [224,225,226]. During conditions of cardiovascular stress including I/R and heart failure, autophagy is also activated [227], however, with potential benefits during ischemia and detrimental effects during reperfusion [228]. In an animal model, the loss of autophagy induced the accumulation of autophagic material in many tissues, and in striated myocytes depressed cardiac contractile function, thus triggering cardiac dysfunction and heart failure [229]. Although autophagy during energy starvation is generally protective [230], excessive autophagy has been associated with heart failure [231,232], myocardial infarction, atrial fibrillation [233], and cardiomyocyte atrophy [234].

According to various studies, it has been found that multiple proteins serve as bridges between autophagy and the NRF2 pathway [235,236,237], for example, by p62 control of KEAP1 turnover by assembling into autophagosomes the LC3-p62-KEAP1 complex, which is eliminated by selective autophagy. The degradation of KEAP1 enables NRF2 to activate and enhance antioxidant defenses [237,238]. Likewise, in atherosclerosis animal models, the NRF2 pathway exhibits both pro- and anti-atherogenic effects during atherosclerosis development. Protection is conferred through the induction of anti-oxidative genes, decreasing inflammatory gene expression and lipid peroxide production in the early stage [239,240]. A proatherogenic role of NRF2 signaling in the advanced stage of atherosclerotic plaque formation has been reported [241] (e.g., in foam cell formation [242], macrophage polarization [243,244], and inflammasome activation). Therefore, the noncanonical NRF2 pathway, which is activated by p62, may represent a dual-target therapy for preventing and reducing CVD.

The induction of antioxidant and detoxification enzymes is known to offer protection for the heart, but the activation of NRF2 can provide even more benefits. These include enhancing anabolic metabolism, promoting autophagy, aiding in the healing of wounds, and suppressing inflammatory responses. In mouse or rat models of myocardial infarction, a number of chemical entities have been tested as NRF2 activators for their potential to reduce infarct size, preserve left ventricular function, and decrease adverse events such as arrhythmia [245,246,247]. Recent research suggests that caloric restriction may have a protective effect in terms of both preventing and slowing down the progression of cardiovascular disease [248]. During caloric restriction, the activation of the NRF2/ARE pathway is important for antioxidant protection but also for maintaining metabolic homeostasis during vascular aging [249]. Likewise, exercise has a positive effect on proteostasis, autophagy, mitochondrial function, and indirectly, proteasome function [249,250,251], which results in angiogenesis, OS, and inflammation amelioration [252,253]. Moreover, exercise-induced cardiac hypertrophy preconditioning enhances resistance to pathological stress through anti-hypertrophic effects mediated by the Mhrt779/Brg1/Hdac2/p-Akt/p-GSK3β signaling pathway [254], and this anti-fibrotic effect is dependent on the NRF2 signaling pathway [255]. The activation results in the increased expression of NRF2-target genes *HO-1*, *NQO-1*, and *GCLC*, leading to the reduced expression of OS and fibrosis-related genes. The overexpression of NRF2, specifically in the cardiac cells of mice, has been shown to have a beneficial effect on the heart (e.g., by reduced infarct size, fibrosis, inflammation and OS), which is also mediated by the translocation of NRF2 into the nucleus [256]. Aging and atherosclerosis are associated with impaired proteasomal function [256,257,258]. This decline in proteasome activity leads to the accumulation of oxidized proteins, which clump together to form large, dysfunctional aggregates. These aggregates become heavily crosslinked and further modified by AGEs, lipid peroxides like HNE, and ubiquitin, preventing unfolding and proteasomal degradation. Furthermore, these protein aggregates can directly inhibit further proteasomal activity, creating a vicious cycle of cellular dysfunction.

Recent evidence suggests a reciprocal regulatory loop between NRF2 and the 26S proteasome. Under homeostatic conditions, the 26S proteasome maintains basal NRF2 levels through ubiquitination and subsequent degradation. However, upon exposure to cellular stress, NRF2 escapes proteasomal degradation and translocates to the nucleus, where it upregulates the expression of genes encoding proteasomal subunits. This NRF2-mediated induction of proteasomal machinery potentially enhances the cell’s capacity for proteostasis, facilitating the clearance of oxidized proteins [259].

Enhancement of NRF2 signaling (e.g., by food-derived compounds such as SFN) could offer greater protection for blood vessels by both detoxifying ROS and promoting the removal of damaged proteins, increasing protection against sustained OS [260]. Exposure of mice to traffic-derived nanoparticles (ultrafine particulate matter) significantly increased the proteolytic capacity in the lung, liver, and heart, which was also evident by an up to twofold increase in the expression of the 20S proteasome, also called the immunoproteasome, and NRF2 [261]. The authors interpreted the exacerbated proteostatic activity as a response to oxidative damage caused by air pollution, as previously reported [262], which declined with the age of the mice (e.g., through upregulation of the NRF2 transcriptional inhibitors BACH1 and c-MYC). Accordingly, pharmacological NRF2 activation may promote efficient proteasomal degradation of damaged proteins in response to air pollution, which may be of specific importance in older individuals with an impaired NRF2 stress response.

In Drosophila, NRF2 mediated the FOXO overexpression-dependent activation of the UPS to counteract organismal age-related defects as well as in cardiomyocytes [263].

The proteostasis network and KEAP1/NRF2 pathway, influencing each other and cooperating to protect cells from damage, have emerged as appealing targets for therapeutic interventions in CVDs. Understanding the mechanisms that govern proteostasis in the context of CVDs may offer potential therapeutic targets for the management of these diseases.

Despite numerous positive reports on the beneficial cardiovascular effects of NRF2 activator therapy, there are also examples of clinical drawbacks. CDDO-methyl ester (CDDO-Me, RTA 402, NSC 713200) was the first CDDO that reached in clinical trials for the treatment of diabetic nephropathy [264]. Although the results of the phase II trial were very encouraging, CDDO-Me was later withdrawn at phase III (BEACON trial) due to cardiovascular safety issues [265] that were not related to NRF2 but most likely to an off-target alteration of endothelin signaling [266,267].

### 4.3. Metabolic Diseases (Obesity, Type 2 Diabetes Mellitus)

The WHO 2023 report on healthy aging highlights unhealthy diets, insufficient physical activity, and sedentary behaviors as widespread factors contributing to the growing burden of chronic illnesses among older adults (https://www.who.int/europe/publications/i/item/WHO-EURO-2023-8002-47770-70520, accessed on). In this regard, the metabolic shift that has occurred in Westernized and developing countries, namely “nutrition transition” (i.e., the consumption of ultra-processed foods, high in calorie intake and poor in nutrients, associated with the increase in sedentary lifestyle) has resulted in a dramatic rise in the prevalence of metabolic diseases [268] that, on a cellular scale, can be defined as alterations in the ability to convert food into energy. It is estimated that one third of the population worldwide is overweight or obese, and that nearly 10% of the world’s adult population (422 million) live with diabetes. It is also estimated that one in three adult Americans has metabolic syndrome (MetS) [269,270,271], a cluster of risk factors for cardiovascular diseases associated with obesity, insulin resistance, and dyslipidemia. Insulin resistance (IR) manifests with impaired fasting glucose and/or impaired glucose tolerance. The progression of IR to overt T2D can be slowed or reversed by lifestyle changes or medications that improve insulin sensitivity or reduce glucose production by the liver. The mechanisms by which T2D develops are not fully understood [271], nevertheless, regardless of the causative factors, the key driver of diabetic phenotypes including the development of IR and dyslipidemia is a propathogenic shift in cellular metabolism. Preclinical studies modeling MetS, such as those deriving from high-fat or Western diet feeding, have shown that rodents displaying metabolic alterations are also characterized by an increased inflammatory state and OS both in the periphery and central nervous system [272,273,274,275], thus strengthening the body of evidence also relating metabolic stressors and unbalanced nutrition to the onset of age-related neurodegenerative disorders [276]. On the other hand, obesity, being intimately connected with IR, may lead not only to the development of T2D and MetS, but also to metabolic-associated steatotic liver disease (MASLD, previously designated as NAFLD) [269,271], a disorder characterized by increased inflammation and OS.

Proteotoxic stress is a key mediator of several diabetic outcomes, thus playing a critical role in the pathogenesis of metabolic diseases [277,278]. One critical regulator of cellular health, particularly during stress, is NRF2, which has been shown to play a role in the regulation of intracellular proteolytic degradation systems (i.e., the proteasome and autophagy-lysosome pathway) as well as molecular chaperones designed to maintain proper protein conformation and stability [68]. For example, studies in rats have provided evidence for a female-specific vulnerability to MASLD in response to a long administration of a high-fat diet supplemented with liquid fructose [279,280]. In particular, the phenotypic manifestation of MASLD in females has been associated with an alteration of the KEAP1/NRF2 axis in the hepatocytes, with increased levels of KEAP1 inhibitor and reduced NRF2 expression in the nuclear compartment. The downstream effects resulted in reduced levels of the antioxidant enzyme HO-1 in female rats compared with males associated with impairment in the autophagic flux, overall suggesting that liver damage in females might derive from dysfunction in the proteostatic process [281]. A different animal model of age-related dysmetabolic alterations has shown intriguing sex-differences regarding the outcomes of a Western diet regimen. In particular, old male mice were characterized by more severe clinical manifestations of MetS (increased IR and dyslipidemia), while females were more vulnerable to central effects of an unbalanced nutrition [282]. The administration of the polyphenol RA was able to prevent metabolic disruptions in males and improve the cognitive abilities in both males and females [282]. Although in this study the role of NRF2 was not specifically investigated, a wider proteomic approach revealed significant RA-dependent anti-aging effects dealing with changes in OS pathways particularly in males (GSH) as well as altered immune- and sex hormone-related signaling pathways in both males and females [282]. Thus, the issue of sex-differences in the context of aging, OS, and NRF2-dependent proteostatic control is particularly worthy of investigation and will be further treated.

As far as specific mechanisms are concerned, it has been shown that impaired protein degradation by the proteasome may play an important role in the development of age-related metabolic diseases such as T2D. NRF2, in particular, has been demonstrated to regulate several proteasomal subunits, especially during OS. Indeed, it has been proven that mice treated with the indirect antioxidant, dithiolethione, enhanced the expression of the 26S proteasome subunits (Psma1, Psma7, Psmb3, Psmb5 and ARE, and Psmb6) in the liver. In contrast, no increase was observed in mice where the transcription factor NRF2 was disrupted. NRF2 has also been shown to bind to POMP, which is critical for proteasome assembly [148,278]. Overall, it is suggested that NRF2 activation is the key in proteasome assembly, while the induction of the 26S proteasome through the NRF2 pathway represents an important indirect action of these antioxidants that can contribute to their protective effects against chronic diseases [68,148,278].

## 5. Design of Targeted Molecules and Innovative Drugs

The identification of molecular targets aimed at developing tailored therapeutic interventions to extend “healthspan” is the current challenge of the scientific community involved in aging research. Given the role played by OS in aging and aging-related diseases, the development of drugs able to modulate NRF2 has gained significant interest.

Currently, two NRF2 activators have been approved for clinical use. (1) Dimethyl fumarate (DMF, BG-12, Tecfidera^®^, Biogen Cambridge, MA, USA), prescribed for relapsing multiple sclerosis and omaveloxolone, which received FDA approval in February 2023 for the treatment of Friedreich’s ataxia, a rare inherited neurodegenerative disorder [283]. DMF exerts its effects by covalently and non-specifically modifying protein nucleophilic groups, such as cysteine thiols, leading to the activation of traditional NRF2 targets like NQO1 and glutathione transferases [284]. However, the non-specific reactivity of DMF may result in a decreased lymphocyte counts, increasing the risk of brain infection [285,286]. (2) Monomethyl fumarate (MMF) and monoethyl fumarate (MEF) are structurally related compounds to DMF. Due to their reduced electrophilicity compared to DMF, recent research has shifted toward these congeners to develop safer medications. MEF, for instance, primarily targets the Cys151 site on KEAP1 and shows considerably less reactivity with other KEAP1 cysteines compared to DMF [287]. Diroximel fumarate, an MMF prodrug developed by Alkermes, has shown fewer side effects and is currently undergoing phase III clinical trials for multiple sclerosis (NCT03093324). Another MMF prodrug, tepilamide fumarate (XP23829), which offers better solubility, permeability, and potency with fewer adverse effects, is in phase II clinical trials for plaque psoriasis (NCT02173301). Omaveloxolone is thought to activate NRF2 mainly through the inhibition of KEAP1, and addresses energy deficits linked to mitochondrial dysfunction. Furthermore, long-term mitochondrial stabilization, facilitated by NRF2-driven transcriptional reprogramming, could contribute to its cytoprotective effects. It is important to note that the long-term safety profiles of NRF2 activators including DMF and its derivatives are not yet fully understood due to ongoing trials. Emerging drug-like compounds are being developed that can directly interfere with the KEAP1/NRF2 interaction or exert similar effects. According to Clinicaltrials.gov, nearly 100 clinical trials are currently investigating NRF2 activators, which include small molecules, dietary supplements, or natural products. Notably, at least 32 clinical trials including chronic illnesses such as cancer, autism, chronic kidney disease, type 2 diabetes, and many more have employed SFN [288]. SFN, a pharmacological NRF2 activator, has been shown to upregulate autophagy and proteasomal genes [289]. It is interesting to note that SFN elevated Beclin-1 and LC3-II, indicating that the activation of NRF2 may aid in the autophagic clearance of misfolded proteins [5]. Additionally, SFN has been linked to enhanced proteasomal degradation, demonstrated by the increased breakdown of mutant huntingtin protein (mHtt) [290]. A recent strategy for NRF2 activation uses pro-drug versions of pro-electrophilic drugs (PEDs), like CA, an active compound from rosemary (*Rosmarinus officinalis*) [291,292,293]. Other notable PEDs include zonarol and isozonarol, derived from the seaweed *Dictyopteris undulata* [287]. Unlike conventional electrophiles, there is no direct reaction between these PEDs and cysteine thiols. Nevertheless, OS forces them to change from hydroquinone to quinone, which is an active electrophile. The subsequent transcriptional activity of KEAP1/NRF2/ARE is triggered by these electrophiles and results in the synthesis of phase II enzymes that are anti-inflammatory and antioxidant [294]. PEDs do not randomly react with other thiols, like GSH, because they are not activated in normal cells. Moreover, cells experiencing OS typically have lower GSH levels, reducing the likelihood of these electrophiles reacting with GSH. As a result, the electrophiles produced by the PED do not come into contact with the typically high levels of GSH with which to react [294]. Furthermore, PEDs have a longer-lasting and stronger effect than conventional antioxidant drugs because they activate a transcriptional pathway that produces endogenous antioxidant enzymes [291,292,293].

Peptides that can block the KEAP1-NRF2 protein-protein interaction (PPI) have been shown to offer protection in disease models [295]. Non-covalent inhibitors of the KEAP1–NRF2 interaction target NRF2’s DLG and ETGE motifs, which bind to KEAP1’s Kelch domain [296]. High-throughput screening using peptide displacement has allowed for the identification of small compounds that obstruct this KEAP1-NRF2 binding [296]. As a result, KEAP1-NRF2 PPI inhibitors are being actively explored as NRF2 activators across various models. While most studies have focused on inhibiting the KEAP1–NRF2 interaction via the ETGE motif, this binding interaction has an extremely high affinity, making it difficult to disrupt [296]. The p62 STGE motif presents another promising target, as it can compete with the NRF2 ETGE motif for binding to KEAP1 [297]. On the basis of this Neh2 ETGE domain, Colarusso et al. identified critical sequences from a pool of linear and cyclic peptide inhibitors for the KEAP1/NRF2 PPI [298]. In a recent study, a synthetic, lipophilic polymer backbone with a water-soluble polymer chain in which each monomer unit contains peptides that bind to KEAP1 makes up a protein-like polymer (PLP) was used [299]. This PLP reportedly inhibits the KEAP1-NRF2 PPI with a high affinity for KEAP1. While long-term safety is still being assessed, the FDA recently approved omaveloxolone, another small molecule, for the treatment of Friedreich’s ataxia by inhibiting the KEAP1/NRF2 PPI network [300]. However, many prototype compounds identified through in vitro studies exhibit poor characteristics related to absorption, metabolism, distribution, and excretion (ADME). Developing PPI inhibitors with suitable drug metabolism and pharmacokinetic properties remains a significant challenge, as long-term NRF2 activation with a strong safety profile is essential. C4X Discovery and Keapstone are also working on novel PPI inhibitors, using structure-based drug design, computational chemistry, and protein crystallography to address these issues (Table 1).

Natural antioxidants may help in treating NRF2-dependent inflammatory diseases (Table 2). Curcumin is a compound from polyphenols that helps reduce inflammation. It does this by increasing the expression of HO-1 and activating the NRF2 pathway [301]. Bisdemethoxycurcumin, a curcumin analog, has been shown to induce HO-1 expression via the NRF2 pathway in LPS-stimulated macrophages [302]. Moreover, in β-cells, demethoxy curcuminoids activate NRF2 through the PI3K/Akt pathway [303]. Tiliroside, a glycoside containing kaempferol, increases the nuclear NRF2 levels and promotes the expression of HO-1 and NQO1, thereby enhancing the antioxidant capacity of various cell types [304]. An additional experiment revealed that engeletin, also known as dihydrokaempferol 3-rhamnoside, enhanced the antioxidant capacity, reduced the release of proinflammatory cytokines, and showed an upregulation of NRF2 and a downregulation of KEAP1 [304]. In chronic arthritis, quercetin has been shown to activate phase II enzymes HO-1 as well as NRF2 [305]. Under amyloid-induced OS, rosmarinic acid activates NRF2 via the Akt/GSK-3/Fyn pathway [117]. Kaurenoic acid, a diterpene, reportedly activates the NRF2 pathway and upregulates the expression of NQO1 and HO-1 in models of acute lung injury [306]. Despite the promising effects of these natural compounds, none have yet achieved practical clinical application. Factors such as solubility, cell permeability, and protein binding, along with the concentration and distribution of NRF2-activating drugs in target tissues, are critical in determining the degree of NRF2 activation, which ultimately affects the desired long-lasting pharmacodynamic outcomes. Growing evidence suggests that drugs or molecules able to activate NRF2 might reduce OS, improve proteostasis, and slow cellular senescence. However, it is worth mentioning that prolonged or excessive activation of NRF2 might also be deleterious, leading to paradoxical outcomes such as the increased production of ROS through the upregulation of NADPH oxidase (NOX4) activity [307,308]. Another significant concern associated with the use of current NRF2 activators as pharmacological agents is the potential for cytotoxicity due to off-target effects. These unintended interactions may disrupt cellular processes unrelated to NRF2 activation, potentially leading to adverse outcomes. This remains a critical focus to improve the safety and therapeutic efficacy, particularly for long-term use in chronic conditions. A primary challenge for future research in designing innovative NRF2 activators lies in developing highly selective compounds with minimized off-target effects. Additionally, defining the optimal therapeutic dose is essential to maintain redox homeostasis while respecting the physiological roles of redox-sensitive pathways. This effort must also account for age-related variations and sex- or gender-specific differences in NRF2 signaling and drug metabolism [309], ensuring tailored and effective therapeutic strategies. Recent research suggests that the dose–response curve for NRF2 activation exhibits a U-shaped pattern, much like the hormesis model associated with vital nutrients [310,311]. This intriguing relationship highlights the potential for optimal NRF2 levels to enhance health benefits, reinforcing the importance of balanced nutrient intake [312]. Mutations in the primary NRF2 repressor, Keap1, or in CUL3, a component of the E3 ubiquitin-ligase complex, can enhance our understanding of NRF2 regulation [310]. Enhancing the NADPH/NADP+ ratio can significantly promote anabolic processes in transformed cells, especially when the NRF2 pathway is overstimulated. This effect highlights the importance of carefully regulating these biochemical mechanisms. Enhanced NRF2 signaling has the potential to guide cancer cells in undergoing metabolic reprogramming. This process can increase the availability of NADPH, which may serve as an effective strategy to promote their proliferation and carcinogenesis [312].

Several studies have highlighted NRF2’s dual role, where excessive activation can be detrimental, particularly in age-related diseases such as cancer [313]. Hyperactivation of NRF2 can facilitate resistance to chemotherapy by neutralizing cytotoxic agents, prompting the exploration of NRF2 inhibitors to counteract these effects. Various compounds including small molecules, natural products, conventional drugs, and peptides have been identified as potential NRF2 inhibitors [314]. Among them, brusatol [315] and ML385 [316] selectively reduce NRF2-driven gene expression, with brusatol enhancing cancer cell sensitivity to OS and chemotherapy. Natural compounds like luteolin [317] and halofuginone [318] are being investigated for their ability to diminish NRF2-mediated chemo- and radio-resistance. Despite their safety profile, natural products often lack specificity due to their complex structures and multi-target effects, complicating their precise therapeutic use. Conventional drugs including glucocorticoids [319] and metformin [320] have also demonstrated NRF2-inhibitory effects but may interact with unintended targets, limiting their application. Peptides targeting the NRF2-MAFG-DNA ternary complex [321] exhibit high specificity and represent a promising approach for NRF2 inhibition. However, their clinical potential is limited by challenges in stability and bioavailability. Despite the promise shown by these compounds, further development is essential to create more selective and potent NRF2 inhibitors with minimized off-target interactions. Advances in compound design and delivery methods will be critical to optimizing their therapeutic efficacy for age-related diseases.

## 6. Conclusions and Future Perspectives

The relationship between proteostasis and NRF2 is pivotal in understanding the aging process. In young healthy cells, NRF2 helps in maintaining proteostasis by promoting the expression of genes involved in OS management, proteasomal degradation, and autophagy. However, as cells age, NRF2 activity declines, weakening the cellular response to oxidative damage and protein misfolding. The diminished NRF2 function accelerates the progression of age-related disorders (e.g., neurodegenerative and cardiovascular diseases), which are driven by OS, inflammation, and proteostasis failure. Therefore, the interplay between proteostasis and NRF2 is critical for determining the rate of cellular aging and the onset of age-associated dysfunctions (Figure 2). Therapeutic strategies that target the NRF2 pathway and restore protein homeostasis hold promise for delaying aging and mitigating age-related diseases. One approach is the activation of NRF2 through small molecules like SFN, which can enhance the expression of NRF2 target genes involved in detoxification, antioxidant production, and proteostasis regulation. Another strategy involves the use of proteostasis regulators, such as chaperones, proteasome or autophagy inducers, to restore protein homeostasis and prevent the aggregation of damaged proteins. Due to the multifactorial nature of aging-related diseases, combination therapies that target both NRF2 activation and proteostasis regulation could provide synergistic benefits. Gene therapy and CRISPR-based approaches targeting the NRF2 pathway or key components of proteostasis may also emerge as future treatments to enhance cellular resilience against aging.

However, the dual role of NRF2 highlights the need for careful therapeutic targeting, and further studies are required to fine-tune these interventions, ensuring their safety and efficacy in promoting healthy aging while avoiding potential side effects.

Therefore, it is critical to develop therapeutics that precisely modulate NRF2 activity without triggering adverse effects, such as uncontrolled cell proliferation, which can arise from excessive NRF2 activation. Similarly, chronic enhancement of proteostasis mechanisms must be carefully controlled to avoid disruptions in normal cellular processes. In this scenario, while NRF2 and proteostasis-based therapies hold great promise, several challenges remain. Indeed, despite preclinical studies that have shown promise in targeting NRF2 and proteostasis in age-related disorders, translating these findings into clinical therapies remains a challenge. Further research is needed to optimize drug delivery, minimize side effects, and understand the long-term effects of NRF2 activation. Nevertheless, the absence of NRF2 would negatively influence the NRF2-KEAP1 pathway, resulting in a diminished defense against oxidative or electrophilic stress. To optimize the activation of this pathway, it is beneficial to identify a balance between a maximally tolerated dose, which serves as a threshold before toxicity occurs, and a biologically effective dose that effectively activates the pathway to combat stress. By carefully determining this balance, we can enhance the pathway’s protective functions. Nevertheless, these approaches represent a promising therapeutic avenue for enhancing cellular resilience during aging and combating major age-related diseases.

## Figures and Tables

**Figure 1 biomolecules-15-00113-f001:**
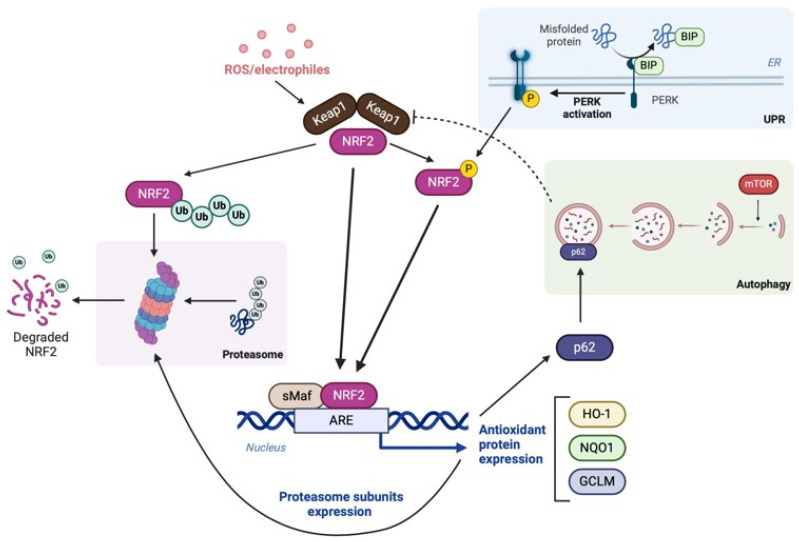
Schematic overview of the NRF2 interaction mechanisms with the unfold protein response (UPR), the mTOR/autophagy pathways and the ubiquitin-proteasome system (UPS). See details in the text (created with BioRender, Toronto, ON, Canada).

**Figure 2 biomolecules-15-00113-f002:**
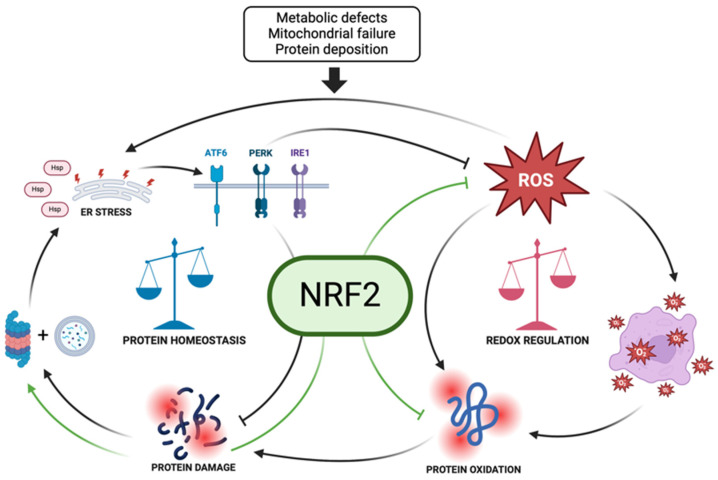
Schematic overview of the central role of NRF2 in regulating protein homeostasis and redox balance. Green lines describe the interventions of NRF2 in homeostatic mechanisms (created with BioRender, Toronto, ON, Canada).

**Table 1 biomolecules-15-00113-t001:** NRF2-targeted therapeutics: approved drugs and compounds in clinical trial stages, along with their mechanisms and clinical applications. The table lists various drugs known to activate NRF2 through different mechanisms, specifically targeting the KEAP1-NRF2 pathway.

**NRF2 Activators**	**Drug Name**		**Mechanism of Action**	**Clinical Use**	**References**
	Tecfidera	Dimethyl fumarate (DMF)	Electrophilic modification of KEAP1-Cys-151	Multiple sclerosis	[284]
	Diroximel fumarate	Monomethyl fumarate (MMF)	Electrophilic modification of KEAP1-Cys-151	Multiple sclerosis (phase III clinical trials)	NCT03093324
	Tepilamide fumarate	Monomethyl fumarate (MMF)	Electrophilic modification of KEAP1-Cys-151	Plaque psoriasis (phase II clinical trials)	NCT02173301
	Sulforaphane (SFN)	1-isothiocyanato-4-(methylsulfonyl)-butane	Electrophilic modification of KEAP1-Cys-151	Cancer, autism, chronic kidney disease, and type 2 diabetes	[288]
	Benfotiamine	Synthetic thiamine precursor	Direct actions on multiple metabolic enzymes, inflammation and oxidative stress	Alzheimer’s disease	NCT02292238
	Carnosic Acid (CA)	Active ingredient in the herb rosemary (*Rosmarinus officinalis*)	Pro-electrophilic drug with antioxidative and ant inflammatory effects	Alzheimer’s disease	[118,291,292,293]
	Zonarol and Isozonarol	Found in seaweed (*Dictyopteris undulata*)	Pro-electrophilic drug with antioxidative and ant inflammatory effects	Alzheimer’s disease	[287]
	Omaveloxolone	KEAP1-NRF2 protein–protein interaction inhibitors	DLG and ETGE motifs on NRF2, which bind to the Kelch domain of KEAP1, are primary targets for non-covalent inhibitors of the KEAP1-NRF2 PPI	Friedreich’s ataxia	[300]
	KEAP1-NRF2 PPI inhibitors		DLG and ETGE motifs on NRF2, which bind to the Kelch domain of KEAP1, are primary targets for the KEAP1-NRF2 PPI	Neurodegenerative disease	[295]

**Table 2 biomolecules-15-00113-t002:** Natural antioxidant compounds with NRF2-activating properties: their mechanisms, applications, and challenges.

**NRF2 Activators**	**Natural Antioxidant Compounds**		**Mechanism of Action**	**Experimental Models**	**References**
	Curcumin	Polyphenol-derived compound	Anti-inflammatory effects	Renal epithelial cells, lung mesenchymal stem cells, macrophages, Leydig cells	[301]
	Bisdemethoxycurcumin	Curcumin analogue	Anti-inflammatory effects	Macrophages, β-cells	[302,303]
	Tiliroside	A glycoside containing kaempferol	Anti-inflammatory and antioxidant effects	Neurons (HT-22 cell line) and microglia (BV22 cell line)	[304]
	Engeletin	Dihydrokaempferol 3-rhamnoside	Anti-inflammatory and antioxidant effects	Microglia (BV22 cell line)	[304]
	Quercetin		Anti-inflammatory and antioxidant effects	Microglia (BV22 cell line) and mouse model for chronic pain	[117,305]
	Kaurenoic acid	Ent-kaur-16-en-19-oic acid (diterpene)	Anti-inflammatory effects	Mouse model for sepsis and with a chronic lung inflammation	[306]

## Data Availability

No new data were created or analyzed in this study.

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
