# Peer review of "Proteostasis Decline and Redox Imbalance in Age-Related Diseases: The Therapeutic Potential of NRF2"

_biomolecules, 2025, doi:10.3390/biom15010113_

Round 1
Reviewer 1 Report
Comments and Suggestions for Authors
This review article on the therapeutic potential of NRF2 against proteostasis decline and redox imbalance in age-related disease has been written by experts in the field within the frame of a EU sponsored COST action. Overall the review is quite interesting but it may be to broad referring to as many as 353 references. It is also felt that some parts are rather repetitive and futher polishing would be recommended. A list of abbreviations is apparently missing.
Author Response
Point-by-point response to Comments from Reviewer 1
Comment 1: This review article on the therapeutic potential of NRF2 against proteostasis decline and redox imbalance in age-related disease has been written by experts in the field within the frame of a EU sponsored COST action. Overall the review is quite interesting but it may be to broad referring to as many as 353 references. It is also felt that some parts are rather repetitive and futher polishing would be recommended. A list of abbreviations is apparently missing.
Response 1: We appreciate the reviewer's insightful feedback regarding the manuscript, and we agree that it required some refinement. Thank you for bringing this to our attention.
In response, we have carefully revised the manuscript to enhance its coherence and readability. Redundancies have been minimized, and the sections identified as repetitive have been restructured to ensure clarity and conciseness. All section has been revised by original authors and reduced for text length and number of references. Additionally, we have included a comprehensive list of abbreviations
Reviewer 2 Report
Comments and Suggestions for Authors
The manuscript provides a good overview of the interactions of Nrf2, redox imbalance and proteostasis. The text is written well. More than 350 references are cited. However, parts of the text and figures should be more precise.
Main points:
Line 38 “As we age, the efficiency of the NRF2 pathway declines” – Please explain why it declines. Is the mechanism known?
Figure 1: the arrow from p62 and autophagy to Nrf2 is not explained well. How can a degradation process increase the amount of Nrf2 or activate it? Maybe a mechanism is mentioned in line 294 (“depends on KEAP1 degradation by p62/SQSTM1”), but this does not fit to the drawing. It is also difficult to understand what the arrow from ROS/electrophiles to Keap1 means. A reference to consider: Komatsu M, Kurokawa H, Waguri S, Taguchi K, Kobayashi A, Ichimura Y, Sou YS, Ueno I, Sakamoto A, Tong KI, Kim M, Nishito Y, Iemura S, Natsume T, Ueno T, Kominami E, Motohashi H, Tanaka K, Yamamoto M. The selective autophagy substrate p62 activates the stress responsive transcription factor Nrf2 through inactivation of Keap1. Nat Cell Biol. 2010;12(3):213-23. doi: 10.1038/ncb2021.
Figure 2: Why is Nrf2 connected to protein oxidation by both an “inhibition” line and an “activation” arrow?
The possible role of environment should be addressed. Line 1006 “environmental challenges the organism has to face” gives some hint, but how?
Minor points:
Line 146 “managing the expression” – this is an usual phrase
Lines 207, 208 “high ROS levels are associated with OS and disease progression” – this statement needs to specific and requires a reference. Which disease?
Line 553 “NRF2 expression declines during the ageing” What is the evidence for this claim?
Tables 1 and 2: Is the format correct? Tables should have a heading instead of a legend.
Author Response
Point-by-point response to Comments from Reviewer 2
Comment 1: The manuscript provides a good overview of the interactions of Nrf2, redox imbalance and proteostasis. The text is written well. More than 350 references are cited. However, parts of the text and figures should be more precise.
Response 1: We thank the Reviewer for their time and effort in evaluating our manuscript. In response, we have carefully revised the manuscript to enhance its coherence and readability. Redundancies have been minimized, the sections identified as repetitive and figures have been restructured to ensure clarity and conciseness. As previously stated, all section has been revised by original authors and reduced for text length and number of references.
Comment 2: Line 38 “As we age, the efficiency of the NRF2 pathway declines” – Please explain why it declines. Is the mechanism known?
Response 2: The decline in the efficiency of the NRF2 pathway with aging is driven by multiple factors, but the precise molecular mechanisms remain under active investigation. We have rephrased the relevant sentences in the Abstract accordingly.
Comment 3: Figure 1: the arrow from p62 and autophagy to Nrf2 is not explained well. How can a degradation process increase the amount of Nrf2 or activate it? Maybe a mechanism is mentioned in line 294 (“depends on KEAP1 degradation by p62/SQSTM1”), but this does not fit to the drawing. It is also difficult to understand what the arrow from ROS/electrophiles to Keap1 means. A reference to consider: Komatsu M, Kurokawa H, Waguri S, Taguchi K, Kobayashi A, Ichimura Y, Sou YS, Ueno I, Sakamoto A, Tong KI, Kim M, Nishito Y, Iemura S, Natsume T, Ueno T, Kominami E, Motohashi H, Tanaka K, Yamamoto M. The selective autophagy substrate p62 activates the stress responsive transcription factor Nrf2 through inactivation of Keap1. Nat Cell Biol. 2010;12(3):213-23. doi: 10.1038/ncb2021.
Response 3: We corrected figure 1 according to reviewer comment by changing the direct arrow between p62 and NRF2 with an inhibition arrow between p62 and Keap1, as also reported in the text.
Comment 4: Figure 2: Why is Nrf2 connected to protein oxidation by both an “inhibition” line and an “activation” arrow?
Response 4: We apologize for the lack of clarity in the figure. We intend to describe with the black direct arrow that the increase of ROS promote protein oxidation, while the induction of NRF2 (green arrow) inhibits this process. We corrected the figure by excluding the interaction between the black arrow and NRF2.
Comment 5: The possible role of environment should be addressed. Line 1006 “environmental challenges the organism has to face” gives some hint, but how?
Response 5: In response to your suggestion, we have rephrased the sentence in the revised manuscript for greater clarity.
Comment 6: Minor points:
Line 146 “managing the expression” – this is an usual phrase
Lines 207, 208 “high ROS levels are associated with OS and disease progression” – this statement needs to specific and requires a reference. Which disease?
Line 553 “NRF2 expression declines during the ageing” What is the evidence for this claim?
Response 6: In response to your suggestion, we have carefully revised the manuscript to enhance its coherence and readability. The sentences in question have been rephrased, and relevant references have been incorporated to improve clarity.
We have addressed the decline in NRF2 pathway efficiency with aging, noting that it is driven by multiple factors, though the precise molecular mechanisms remain under active investigation. Specifically, we have updated the text to emphasize the age-dependent differential expression of NRF2 and KEAP1, as reported by Gao et al. [PMID: 25370996] and Kopacz et al. [PMID: 32487458].
Comment 7: Tables 1 and 2: Is the format correct? Tables should have a heading instead of a legend.
Response 7: We apologize for the inconvenience, we substituted the legend with the heading.
Reviewer 3 Report
Comments and Suggestions for Authors
Nrf2-ARE signaling and proteostasis in the context of some age-related diseases were the subject of several reviews published within last years.
The paper submitted to Biomolecules is the result of COST action project and as a consequence is written by 13 authors. Such multi - author work is difficult to handle. As an effect the paper as a whole and its specific sections often lack of flow and might be substantially shorten including the list of references, which is too long (353!).
According to the title the aim of this review was the to describe the role of the proteostasis decline and redox imbalance in ageing relative diseases with Nrf2 as therapeutic target.
Consequently, it would better if the description of specific diseases was preceded by the section 4 referring to ageing and longevity as the latter is certainly affected by ageing related diseases.
The critical comment concerning lack of flow particularly is addressed to the section 3.2 (Cardiovascular diseases) which certainly should be shortened. Moreover, although the constitutive activation of Nrf2 in cancer cells is mentioned in this section (lines 633-643), in line 785 clinical draw back of Nrf2 activation therapy are interpreted as not related to Nrf2, while dual function of Nrf2 i.e. exacerbating pathophysiological CVD outcomes is well described.
In general, the dual role of Nrf2 in disorders described in this paper should be better addressed and mentioned in conclusions. Along with the activators of Nfr2 pathway its inhibitors are equally needed and searched.
Moreover, it would be worth to include at least short paragraph addressed the problem of Nrf2 and proteostasis in cancer cells as this disease also more often occurs in elderly people. It might be done in expense of e.g. the description of Down syndrome.
Minor remarks
The statements “activation of proteostasis” line 69 or “enhance proteostasis” should be rather avoided as proteostasis means protein homeostasis.
Author Response
Point-by-point response to Comments from Reviewer 3
Comment 1: Nrf2-ARE signaling and proteostasis in the context of some age-related diseases were the subject of several reviews published within last years.
The paper submitted to Biomolecules is the result of COST action project and as a consequence is written by 13 authors. Such multi - author work is difficult to handle. As an effect the paper as a whole and its specific sections often lack of flow and might be substantially shorten including the list of references, which is too long (353!).
According to the title the aim of this review was the to describe the role of the proteostasis decline and redox imbalance in ageing relative diseases with Nrf2 as therapeutic target.
Response 1: Thank you for your detailed feedback on our manuscript.
We have carefully revised the manuscript to enhance its coherence and readability. Redundancies have been minimized, the sections identified as repetitive and figures have been restructured to ensure clarity and conciseness. All section has been revised by original authors and reduced for text length and number of references.
Comment 2: Consequently, it would better if the description of specific diseases was preceded by the section 4 referring to ageing and longevity as the latter is certainly affected by ageing related diseases.
Response 2: We agree that discussing aging and longevity before specific age-related diseases provides a more logical flow and context for the review. We re-organized the section according to this suggestion
Comment 3: The critical comment concerning lack of flow particularly is addressed to the section 3.2 (Cardiovascular diseases) which certainly should be shortened. Moreover, although the constitutive activation of Nrf2 in cancer cells is mentioned in this section (lines 633-643), in line 785 clinical draw back of Nrf2 activation therapy are interpreted as not related to Nrf2, while dual function of Nrf2 i.e. exacerbating pathophysiological CVD outcomes is well described.
Response 3: We thank the reviewer for his helpful comments and in accordance with his suggestions we have revised section 3.2 by reducing its length. We have also eliminated the part concerning the negative role of NRF2 associated with tumors.
Comment 4: In general, the dual role of Nrf2 in disorders described in this paper should be better addressed and mentioned in conclusions. Along with the activators of Nfr2 pathway its inhibitors are equally needed and searched.
Response 4: We have addressed the dual role of NRF2 in the disorders discussed in the paper and have added relevant information on both NRF2 activators and inhibitors in the revised manuscript.
Comment 5: Moreover, it would be worth to include at least short paragraph addressed the problem of Nrf2 and proteostasis in cancer cells as this disease also more often occurs in elderly people. It might be done in expense of e.g. the description of Down syndrome.
Response 5: We added a paragraph concerning Cancer development in Down syndrome and the potential role of NRF2
Comment 6: Minor remarks
The statements “activation of proteostasis” line 69 or “enhance proteostasis” should be rather avoided as proteostasis means protein homeostasis.
Response 6: We apologize for the oversight and have rephrased the relevant sentences to address this inaccuracy.
Round 2
Reviewer 3 Report
Comments and Suggestions for Authors
Basically, the authors responded correctly to my critical comments, although the paper is now longer than original version. The number of references was only slightly reduced (353 vs 320).
The authors have misunderstood my remark # 5. I have suggested to include paragraph concerning Nrf2 and cancer instead of long description of Down Syndrome not specifically in this disorder. However, I can accept it.
The decision concerning the volume of the paper and the number of references I leave to Editors.